# Etiology and Diagnosis of Permanent Hypoparathyroidism after Total Thyroidectomy

**DOI:** 10.3390/jcm10030543

**Published:** 2021-02-02

**Authors:** Antonio Sitges-Serra

**Affiliations:** Endocrine Surgery, Hospital del Mar, 08003 Barcelona, Spain; assmolinos@gmail.com; Tel.: +34-649-191-546

**Keywords:** thyroidectomy, parathyroid failure, PGRIS, splinting, recovery, hypoparathyroidism

## Abstract

Postoperative parathyroid failure is the commonest adverse effect of total thyroidectomy, which is a widely used surgical procedure to treat both benign and malignant thyroid disorders. The present review focuses on the scientific gap and lack of data regarding the time period elapsed between the immediate postoperative period, when hypocalcemia is usually detected by the surgeon, and permanent hypoparathyroidism often seen by an endocrinologist months or years later. Parathyroid failure after thyroidectomy results from a combination of trauma, devascularization, inadvertent resection, and/or autotransplantation, all resulting in an early drop of iPTH (intact parathyroid hormone) requiring replacement therapy with calcium and calcitriol. There is very little or no role for other factors such as vitamin D deficiency, calcitonin, or magnesium. Recovery of the parathyroid function is a dynamic process evolving over months and cannot be predicted on the basis of early serum calcium and iPTH measurements; it depends on the number of parathyroid glands remaining in situ (PGRIS)—not autotransplanted nor inadvertently excised—and on early administration of full-dose replacement therapy to avoid hypocalcemia during the first days/weeks after thyroidectomy.


*A clever person solves a problem.*

*A wise person avoids it.*
Albert Einstein

## 1. Introduction

Between 75% and 85% of all cases of permanent hypoparathyroidism result from neck surgery, mostly from total thyroidectomy, particularly if performed for cancer. Although the biochemical and clinical profiles of permanent hypoparathyroidism are currently well known [1,2], there is some lack of knowledge on the causes of postoperative parathyroid insufficiency and the dynamics of hypoparathyroidism from the early postoperative stages to recovery of the parathyroid function or, else, to permanent dependency on replacement therapy.

Risk factors for parathyroid failure after total thyroidectomy, such as Graves’ disease, lymph node dissection, or case load have been identified in recent analysis [3,4] and will not be the subject of the present review. In one way or another, these result in some kind of injury to the parathyroid glands (fragmentation, thermal injury, devascularization, inadvertent resection, autotransplantation into the sternocleidomastoid muscle). The only exception might be the higher prevalence of hypoparathyroidism syndromes in young women in whom, besides an iPTH drop, other endocrine regulators such as the estrogens may play a role, particularly in total thyroidectomy for benign disease [5]. 

This narrative review introduces some new concepts concerning the physiological events that take place between the knocking out of the parathyroid function after thyroidectomy and its eventual recovery along weeks or months, and challenges the diagnosis of permanent hypoparathyroidism being made before two years after surgery [6,7]. It is based on a subjective appreciation of articles, mostly of the observational type, that have substantially contributed to the understanding of the “events within a black box” occurring between surgery and permanent hypoparathyroidism.

## 2. Definition of Stages of an Evolving Iatrogenic Disease

Several terms have been employed in the literature to describe postoperative parathyroid insufficiency, but none has reached wide consensus nor has incorporated the time dimension [8]. Thus, before analyzing the physiological events that take place during the different phases of postoperative parathyroid dysfunction, we make the following definitions proposal:

*Post-surgical or postoperative hypoparathyroidism.* A generic definition encompassing all syndromes of short-, medium-, and long-term parathyroid failure after neck surgery.

*Postoperative parathyroid failure*. This term should be preferred to others commonly used to describe the scenario of patients presenting symptomatic or asymptomatic (biochemical) hypocalcemia and requiring replacement therapy at the time of hospital discharge, such as transient hypoparathyroidism, postoperative hypocalcemia, or postoperative hypoparathyroidism. “Transient hypoparathyroidism” should only be employed after recovery of the parathyroid function, since the “transient” nature of hypoparathyroidism cannot be predicted beforehand. In addition, “transient” does not imply a given period of time, since recovery of the parathyroid function may take days, months, or even more than a year. “Postoperative hypocalcemia” was much used before early measurements of iPTH serum concentrations were available, since it was the “phenotype” of the iPTH decline occurring after surgery. Calcium was measured as a surrogate and replacement therapy started if concentrations were <8 mg/dL (2 mmol/L) and/or patients developed clinical symptoms. The difficulties of using this term are that the time interval between surgery and blood testing may over- or infradiagnose hypocalcemia and that different cut-off values for serum calcium (s-Ca) concentrations have been used to define “hypocalcemia” [9]. Currently, no patient should be allowed to develop post-thyroidectomy hypocalcemia (neither biochemical nor symptomatic), and there is a growing consensus that replacement therapy should be started if iPTH concentrations drop below 10–15 pg/mL or decline more than 70% of the preoperative values at four hours after surgery [10,11,12,13], thus making obsolete a syndrome definition based on s-Ca measurements.

*Protracted hypoparathyroidism* is defined as the need for replacement therapy beyond one month after neck surgery due to persistently low or even absent circulating iPTH concentrations. It is a clinically relevant syndrome, since it includes a group of patients at high risk (25%) for developing permanent hypoparathyroidism [14]. Furthermore, at this time point, a first prediction can be made of the probabilities of parathyroid function recovery in a given patient based on s-Ca and iPTH measurements (Figure 1) [15]. This can orientate the information given to an anxious patient on prolonged replacement therapy and implies the need for long-term close monitoring.

*Permanent hypoparathyroidism*. This is defined as subnormal or absent iPTH serum concentrations more than one year after thyroidectomy, making the patients dependent on replacement therapy for life. The time limit to consider protracted hypoparathyroidism becoming permanent is under debate. No doubt, six months is too early to diagnose permanent hypoparathyroidism, since only two-thirds of patients with protracted hypoparathyroidism will recover within this time period. Of the remaining patients, most will recover within a year, but up to 10–15% will recover after one-year follow-up. Our group has identified three variables that make recovery of the parathyroid function still possible beyond one year: an s-Ca ≥ 9 mg/dL and detectable iPTH concentrations one month after thyroidectomy, and four parathyroid glands remaining in situ (PGRIS 4) [16]. “Permanent” seems preferable to the seldom used term “chronic” hypoparathyroidism, since it refers to the absence of recovery of the parathyroid function as opposed to “transient”.

*Partial (subclinical, relative) hypoparathyroidism*. At any time after surgery, some patients may present with episodes of hypocalcemia requiring replacement therapy despite their iPTH concentrations being within the normal range. This is a consequence of both a decreased parathyroid functional reserve due to a reduction of the secreting parenchyma and to comorbidities impairing the absorption of calcium and/or vitamin D from the gastrointestinal tract such as infectious diarrhea, short bowel syndrome, extended right hemicolectomy, treatment with monoclonal antibodies (denosumab), or previous bariatric surgery. It has been reported that after total thyroidectomy, there is a blunted response to hypocalcemia due to a limitation of the secretory capacity of the parathyroid glands, with an inappropriate response to low serum calcium levels [17]. Patients with partial hypoparathyroidism may eventually keep a normal/low s-Ca in basal circumstances but develop symptomatic hypocalcemia when faced with a metabolic challenge.

The flow diagram in Figure 2 shows the different phases of postoperative hypoparathyroidism.

## 3. Postoperative Parathyroid Failure: A Too Common Complication

Post-thyroidectomy tetany and its connection with serum calcium and parathyroid gland injury were recognized as soon as surgeons accomplished thyroid gland resection with low postoperative mortality. After World War II, conservative thyroid procedures were recommended and widely adopted in order to preserve the parathyroid glands [18,19]. However, starting in the early 1980s, fear of goiter recurrence after nodulectomies or subtotal resections, and promotion of radical surgery for cancer, led to the enthusiastic and widespread adoption of total thyroidectomy, resulting in an epidemic of permanent hypoparathyroidism affecting up to 5–15% of total thyroidectomy patients according to multicenter studies [4,20,21,22,23,24]. This was also fuelled by misconceptions about the benefits of parathyroid autotransplantation in the sternocleidomastoid muscle as an effective maneuver to prevent permanent hypoparathyroidism, and the belief that inadvertent parathyroidectomy had a null impact on postoperative parathyroid function (see below). Both should be considered currently as the principal technical pitfalls leading to permanent hypoparathyroidism. The era of total thyroidectomy for everybody protected by parathyroid autotransplantation has reached its end.

Post-thyroidectomy parathyroid failure is currently regarded as the most common complication of total thyroidectomy and the first cause of return to hospital after discharge [25]. The crippling nature of permanent postoperative hypoparathyroidism has been widely recognized [1,2,26,27]. However, surgeons are quite reluctant to accept the idea that parathyroid injury is, essentially, a technical issue. Obviously, there are situations in which parathyroid identification and preservation can be difficult, but this cannot be an argument to ignore lack of experience or underscore appropriate training. Lack of expertise and low volume of surgery are directly responsible for technical mistakes. This has been manifested recently in Denmark, where an unfortunate political decision put thyroid surgery in the hands of inexperienced otorhinolaryngologists, resulting in an epidemic of permanent hypoparathyroidism reaching up to 13.4% at six months just for benign disease [28]. Along with the South Korean papillary thyroid cancer overtreatment epidemic [22,29], the Danish experience shows how deleterious political decisions can be for the population’s health.

## 4. Intraoperative Management of the Parathyroid Glands

Three technical mishaps predispose to transient and permanent postoperative parathyroid failure: inadvertent parathyroidectomy, autotransplantation of normal parathyroid tissue, and devascularization.

*Inadvertent resection*. Accidental removal of normal parathyroid glands has been recognized only recently as a main contributor to post-surgical hypoparathyroidism. Early studies suggested that accidental parathyroidectomy did not result in an increase of hypoparathyroidism rates [30,31,32]. However, most of these initial reports were retrospective and did lack precise definitions, appropriate follow up, or a statistically meaningful number of patients. A consensus has been reached during the last decade that accidental parathyroidecotmy increases the prevalence of permanent hypoparathyroidism up to 5–8% (Table 1) [14,24,33,34,35,36,37,38,39]. In addition, the recent literature largely confirms that lymphadenectomy for thyroid cancer is the main risk factor, particularly when extrathyroidal extension is present.

A recent randomized trial suggests that parathyroid autofluorescence may decrease the rate of incidental parathyroidectomy, but this is the first of its class, which is published by authors without a background research on hypoparathyroidism, biased recruitment, and with a declared conflict of interest [40]. The concept that autofluorescence may identify the parathyroid glands earlier and better than the naked (experienced) eye sounds bizarre. In any case, this and other potential parathyroid identification techniques should not lead to underappreciate the importance of a thorough anatomical knowledge before embarking in thyroid and parathyroid surgery.

A “raise-your-hand survey” was carried out during the British Association of Endocrine and Thyroid Surgeons (BAETS) meeting in Barcelona (October 2019): less than 10% of the 220 surgeons attending did know their own rate of inadvertent parathyroidectomy. It is weird that up to now, no mention has been made in the literature about this as a key parameter to monitor the quality of surgery.

Needles to say, when total thyroidectomy is associated to parathyroidectomy for concomitant primary or secondary hyperparathyroidism, the risk of permanent hypoparathyroidism approaches that of inadvertent single gland excision [21,41]. 

*Parathyroid autotransplantation*. Recommendation of whole gland autotransplantation during thyroidectomy was already mentioned in Halsted and early Mayo Clinic reports. Failure to prove its efficacy [42] and the functional success of autotransplantation of hyperplastic parathyroid tissue [43] led to the popularization of autotransplantation of fragmented normal glands into the sternocleidomastoid muscle based on its supposed but never proven efficacy to prevent permanent hypoparathyroidism [44,45,46]. Some authors even recommended routine autotransplantation of one gland [47,48]. However, despite its widespread use, the benefit of parathyroid autotransplantation is being currently challenged [24,49] even by surgeons that favored it in the past [21,50,51]. Furthermore, both the rate and the criteria for autotransplantation vary widely, since there is no consensus on when to proceed unless the gland is completely devascularized or identified in the thyroidectomy specimen before is sent to the pathology lab. Color change does not help in decision making [52,53]. 

Ethical considerations do not make it possible to assess the real value of autotransplantation in a controlled trial. However, three recent studies have provided data that can be regarded as the closest possible to those obtained from randomized trials. These independent observations reported similar short and long-term parathyroid function outcomes in PGRIS 3 patients in whom the fourth gland was either accidentally resected (and identified by the pathologist) or autotransplanted [54,55,56] (Table 2).

Interestingly, the rate of permanent hypoparathyroidism reported in these three observational studies, when the four glands were preserved in situ, was also very similar, between 1 and 2%. 

A recent meta-analysis supports avoiding parathyroid autotransplantation [57]. Wang et al. reviewed 25 independent studies comprising 10,531 patients concluding that “parathyroid gland autotransplantation is significantly associated with increased risk of postoperative and protracted hypoparathyroidism and the number of transplanted glands is positively correlated with the incidence of postoperative hypoparathyrodism”. 

*Devascularization*. Parathyroid angiography after indocyanine green injection [58] has proved that, at the time of thyroidectomy, blood flow to the parathyroid glands can be compromised. However, the role of angiography in the prevention of hypoparathyroidism is controversial, since there is not much to do after a positive test other than hoping for the blood flow to be restored shortly after surgery. Despite optimistic claims that fluorescence intensity correlates with iPTH levels measured after surgery [58,59], a close review of Lang’s et al. data shows this correlation to be very weak both immediately after surgery (surprisingly!) and on postoperative day 1. Other limitations of this report are that all parathyroid glands were biopsied to ensure proper identification and no data are given on the possible correlation between fluorescence and s-Ca. Although these same groups [58,59] recommend autotransplantation of the glands that appear poorly vascularized, it is our view that this can only worsen the final outcome of the parathyroid function.

## 5. The Importance of *In Situ* Preservation of the Parathyroid Glands

These findings put emphasis on the need for careful intraoperative identification and handling of the parathyroid glands. This is by no means a new proposal, since decades ago, experts in thyroid surgery were already alerting against aggressive management of the glands during total thyroidectomy and recommended minimal dissection and in situ preservation [60,61,62]. What we know now is that PGRIS is a key player in the pathophysiology of postoperative parathyroid failure, which is a key variable influencing the recovery of the parathyroid function, and it has a definite impact on the rate of permanent hypoparathyroidism [24,56,63,64]. In fact, there is a linear relationship between PGRIS and the prevalence of all hypoparathyroidism syndromes, as shown in Table 3.

## 6. Some Technical Hints

*Surgical anatomy of the parathyroid glands*. Parathyroid gland preservation is technically more demanding than avoiding recurrent laryngeal nerve injury. It implies a thorough knowledge of the normal anatomy and its many variants [65]. Crucial anatomical landmarks are the common subcapsular or very adherent position of some superior and inferior glands, the relationship between the lower pair and the thymus, and the tiny final pedicle connecting the glands to the arterial supply, which are mostly dependent on the inferior thyroid artery. Surgeons should be familiar with the anatomical variations secondary to thyroid disease, particularly in reoperative surgery [66] and thyroidectomies for large goiters and cancer requiring central neck lymph node dissection.

Careful mobilization of the upper pole by opening the cricothyroid space and blunt lateral dissection should be the first operative step once the thyroid gland has been exposed: it allows a three-dimensional view of the area where the upper parathyroid gland is located [60]. Recognizing, isolating, and avoiding clamping the thyro-thymic ligament allows for proper preservation of the lower glands. Energy devices should not be used too close to the parathyroid glands. Recent articles deal with technical tips to ensure preservation of the parathyroid glands along with their subtle vascular supply [67,68]. 

*Central neck dissection*. There is strong evidence that total thyroidectomy associated to central neck lymph node dissection for papillary or medullary cancer is the highest risk procedure for permanent hypoparathyroidism, even in experienced hands, particularly if therapeutic (clinically positive nodes) or if performed at both sides [21,36,38,69,70,71]. This is most probably due to a combination of gland devascularization, liberal autotransplantation policies [21], and inadvertent excision. Whenever possible, surgeons should not proceed to central node clearance without prior identification of the ipsilateral parathyroid glands. Appropriate recognition and preservation of the thyro-thymic ligament is crucial to avoid crushing, devascularization, or excision of the inferior parathyroid glands [72]. Furthermore, parathyroid cell nests often present in the upper thymus may play some protective role against permanent hypoparathyroidism [38]. Unilateral (rarely bilateral) thymectomy may be unavoidable during a therapeutic central neck dissection, and this requires paying much attention to the status of the remaining parathyroid glands.

*The role for conservative surgery*. Perhaps the surgical pendulum has moved too far in the direction of total thyroidectomy, while some patients could be treated with less extensive procedures, particularly when dealing with benign disease. The Dunhill procedure should be considered in patients with asymmetrical goiter [73] and in patients with goiter or autoimmune disease if the parathyroid glands have been injured at the first side [74,75]. Dunhill/near total thyroidectomy is extremely safe regarding the parathyroid function and carries a negligible risk of goiter recurrence [73,74,75,76].

## 7. Hypoparathyroidism after Surgery for Hyperparathyroidism

*Primary hyperparathyroidism*. There are scarce data on the prevalence of hypoparathyroidism after isolated parathyroid surgery (not performed with concomitant thyroidectomy). As most cases of primary hyperparathyroidism are due to single adenomas, hypoparathyroidism in this setting is almost restricted to chief-cell hyperplasia treated by subtotal/total resection plus autograft and remedial surgery. Historical series have reported a prevalence of permanent hypoparathyroidism after subtotal resection ranging from 5 to 24% [77,78,79,80,81,82], and routine replacement therapy at the time of hospital discharge seems advisable after this procedure. Postoperative hypoparathyroidism may spontaneously revert years after subtotal parathyroidectomy in MEN 1 patients due to the hyperplasia of supernumerary glands. Total parathyroidectomy plus forearm autograft (either immediate or after cryopreservation) has been almost abandoned for the treatment of primary parathyroid hyperplasia due to graft failure in over 40% of the patients [83,84,85,86]. In redo surgery, highly experienced surgeons have reported 5 to 15% rates of permanent hypoparathyroidism [87,88].

*Secondary hypoparathyroidism*. Subtotal or total parathyroidectomy with forearm autograft are the most commonly performed procedures for severe renal hyperparathyroidism in patients on hemodialysis. A recent consensus paper reviewing a large cohort of randomized studies reported a similar average of 2% hypoparathyroidism rates for both types of surgery [89]. Total parathyroidectomy has been recommended for patients that are not candidates for a renal transplant. In this setting, hypoparathyroidism can be managed satisfactorily, modifying the calcium concentration of the dialysis fluids.

Transient and permanent hypoparathyroidism after parathyroid surgery result from the extreme reduction of functional parathyroid parenchyma, which is due to either a small or devascularized remnant after subtotal resection, multiple parathyroidectomies in recurrent/persistent cases, extensive bilateral dissection, and failure of autografts. Finally, volume/outcome studies have shown that the surgical caseload is a major determinant of persistent hyperparathyroidism and surgical complications including hypoparathyroidism [90,91,92].

## 8. Postoperative Management

Low postoperative 4-h concentrations of iPTH and/or s-Ca (24 h) are the preferred markers to initiate selective replacement therapy with calcium salts and calcitriol at the time of hospital discharge. Routine calcium replacement, which is still used in some centers, will probably wither as more units will guide therapy based on postoperative iPTH measurements [93,94]. However, they are useful to predict neither the duration nor the final outcome of parathyroid failure. After the pathology report becomes available, the PGRIS category (1–2, 3, or 4) should be determined and added to the patient’s records; together with delayed (around 1 month) s-Ca and iPTH measurements, it can orientate regarding the likelihood and timing of parathyroid function recovery [15,16,62]. It is worth emphasizing again that the diagnosis of permanent hypoparathyroidism should not be made too early, probably not before 18–24 months after surgery since between 10 and 15% of protracted hypoparathyroidism cases (usually PGRIS 4) will recover after 12 months and some will even recover after two years [16,24,95].

Repeated analysis of our own data with an increasing number of total thyroidectomies has revealed the importance of keeping a s-Ca in the high–normal range during the first postoperative days/weeks of replacement therapy to increase the likelihood of parathyroid function recovery [14,15,16]. We have hypothesized that hypocalcemia may represent an additional stress for the ischemic and/or reduced parathyroid parenchyma, and that normocalcemia may be supportive to the failing glands [96]. This is not surprising, since other organ failures (kidney, lungs, heart) have been traditionally managed with supportive strategies. However, it is fair to say that no controlled trial has been carried out so far concerning the most appropriate dosage of calcium salts and calcitriol immediately after surgery for those patients developing postoperative parathyroid failure. In our unit, patients with protracted hypoparathyroidism, who recovered the parathyroid function, did receive significantly higher doses of calcium salts (2.75 vs. 2.12 g/day of Ca^++^) and calcitriol (0.63 vs. 0.38 mcg/day) at the time of hospital discharge than those patients developing permanent hypoparathyroidism [14]. To further support an aggressive medical treatment of postoperative parathyroid failure, Bin Saleem et al. [97] have reported that delayed (>90 days) completion of total thyroidectomy (usually performed after the diagnosis of cancer is made in the initially resected lobe) carries a lesser risk of hypoparathyroidism than early (7–90 days) redo surgery. This may be explained by the contralateral normal parathyroid glands splinting the pair potentially injured on the first operated side. Thus, according to the best evidence available, it seems advisable to avoid hypocalcemia immediately after surgery not just to prevent clinical symptoms but also to provide a better chance for parathyroid function recovery.

## Figures and Tables

**Figure 1 jcm-10-00543-f001:**
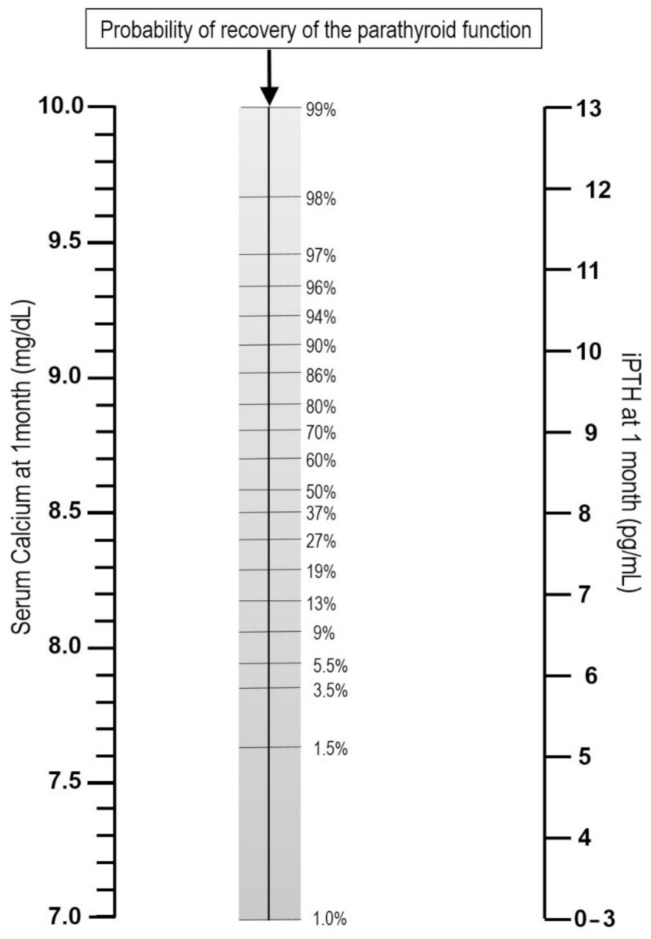
Nomogram to calculate the probability of recovering from protracted hypoparathyroidism based on s-Ca and iPTH (intact parathyroid hormone) measurements one month after total thyroidectomy (Readapted with permission from ref. [15]).

**Figure 2 jcm-10-00543-f002:**
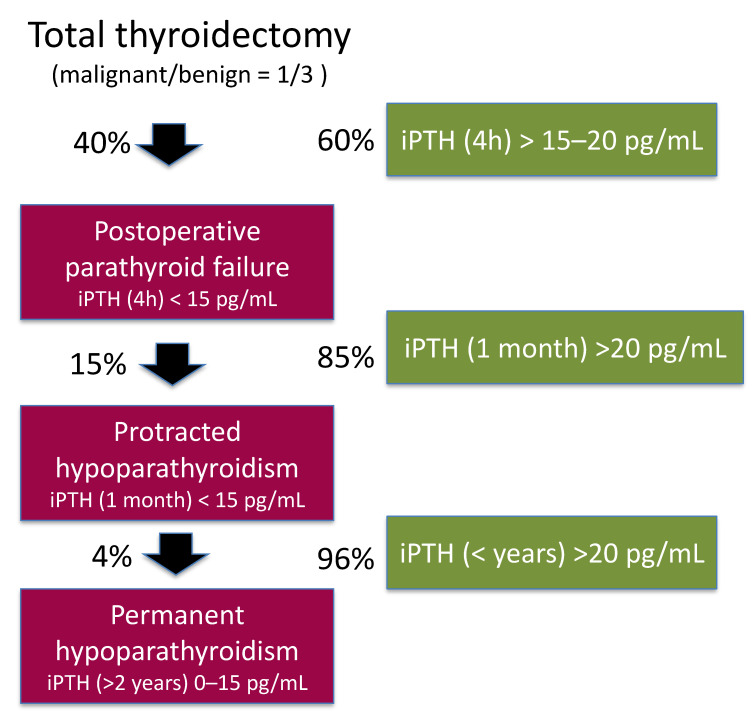
Hypothetical flow diagram showing the different stages of postoperative hypoparathyroidism (red pathway) or recovery of the parathyroid function (green pathway) after 100 total thyroidectomies performed in a high volume and experienced surgical unit (based on refs. [3,4,14,17]).

**Table 1 jcm-10-00543-t001:** Rates of inadvertent parathyroidectomy (%IP) and associated hypoparathyroidism (Transient/Permanent).

1st Author	Ref	%IP	Hypopara T/P	Risk Factors
Rajinikanth, J.	[33]	13%	40%/8.5%	LymphADX
Sorgato, N.	[34]	8%	11%/5%	LymphADX
Sitges-Serra, A.	[14,38]	16%	51%/6.4%	LymphADX Extrathyroidal
Paek, S.H. (PTC)	[35]	29%	34%/6.5%	Extrathyroidal
Applewhite, M.	[36]	16%	T-65%	LymphADX
Zhou, H.	[7]	20%	P-6.7%	LymphADX
Diez, J.J.	[24]	29%	40%/21%	LymphADX

LymphADX = Lymph node dissection. PTC: Only Papillary Thyroid Cancer.

**Table 2 jcm-10-00543-t002:** Rates of postoperative parathyroid failure (PPF) and permanent hypoparathyroidism (PH) after total thyroidectomy when three parathyroid glands were preserved in situ, whereas the fourth one was either autotransplanted or inadvertently resected.

Author (Reference)	1 PG Autotransplanted	1 PG Resected
Lorente-Poch, L., et al. [54]	PPF: 52% PH: 7.3%	PPF: 50% PH: 5.3%
Tartaglia, F., et al. [55]	PPF: 46% PH: 3.5%	PPF: 39% PH: 3.5%
de León-Ballesteros, G.P., et al. [56]	PPF: 29% PH: 3.7%	PPF: 35% PH: 3.9%

**Table 3 jcm-10-00543-t003:** Parathyroid glands preserved in situ (PGRIS) and postoperative parathyroid function after 657 total thyroidectomies (modified from ref. [63]).

Hypoparathyroidism Syndrome	PGRIS 1–2 (*N* = 43)	PGRIS 3 (*N* = 186)	PGRIS 4 (*N* = 428)	*p* Value
Parathyroid failure (*n* = 278) (s-Ca ^24 h^ < 8 mg/dL)	32 (74)	95 (51)	155 (36)	<0.001
Protracted hypoparathyroidism (*n* = 121)	19 (44)	46 (25)	56 (13)	<0.001
Permanent hypoparathyroidism (*n* = 30)	7 (16)	12 (6.5)	11 (2.6)	<0.001
s-Ca postop ^24 h^ (mg/dL)	7.6 ± 0.9	7.9 ± 0.8	8.2 ± 0.8	<0.001
iPTH ^24 h^ (pg/mL)	7.4 ± 7	6.5 ± 8	24.3 ± 21	<0.001
iPTH ^1 month^ (pg/mL)	19.7 ± 23	28 ± 25	37 ± 28	<0.001
Extent of surgery				0.004
TT (*n* = 540)	32 (74)	142 (76)	366 (86)	
TT + CCND (*n* = 63)	9 (21)	25 (13)	29 (7)	
TT + CCND + LCND (*n* = 54)	2 (5)	19 (10)	33 (8)	

s-Ca ^24 h^: serum calcium value 24 h after surgery; iPTH ^24 h^: intact parathyroid hormone at 24 h after surgery; iPTH ^1 month^: intact parathyroid hormone at 1 month after surgery; TT: isolated total thyroidectomy; CCND: central compartment lymph node dissection; LCND: lateral compartment lymph node dissection. Numbers within brackets are percentages.

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
