# Peer review of "Etiology and Diagnosis of Permanent Hypoparathyroidism after Total Thyroidectomy"

_jcm, 2021, doi:10.3390/jcm10030543_

Round 1

Reviewer 1 Report

This article is a narrative review on etiology and diagnosis permanent hypoparathyroidism after total thyroidectomy, presenting some news in this regard. The topic is very interesting, however, in my opinion, the article lacks the methodology. It is not clear the methodology with which the author selected the articles / studies / clinical protocols and for the discussion of some aspects it would be necessary to make a systematic review.

Paragraph: “Definition of stages of an evolving iatrogenic disease”: The author discuss definitions regarding transient and permanent post-surgical hypoparathyroidism. A revision of the definitions, in my opinion, should be carried out as part of a more complex discussion involving more experts in the field.

Paragraph: “Postoperative parathyroid failure: a too common complication”: Other articles regarding the epidemiology of post-surgical hypoparathyroidism should also be analyzed and cited (for example: The Epidemiology of Hypoparathyroidism in Italy: An 8-Year Register-Based Study. Cipriani C, et al. Calcif Tissue Int. 2017 Mar;100(3):278-285.)

Paragraph: “Intraoperative management of the parathyroid glands”: for this discussion, as for others covered in this review, perhaps it would be useful to make a systematic review of the literature or at least the methodology of selection of published articles that has been made should be described.

Author Response

1) This is not a systematic review. It is a narrative review based on the most relevant literature and most importat, on the autos experience and previous publications. It was conceived as "point of view".

2) No changes have been made the definitions since this is an innnovative proposal and there is no current consensus. It is not clear to me the sentence "involving more experts in the field". The manuscript certainly refers to many other authors experience but under a critical viewpoint.

3) I am not deaing with epidemiology of the disease, rather on etiology and diagnosis.

4) Article selection has been made based on the author's opinion as to their relevance. It is by no means an exhaustive literarture review which is out of the reach of this more conceptual revision.

5) The reviewer makes no criticism on the core of the paper which deals with etiology and diagosis issues.

Reviewer 2 Report

The manuscript is well designed and structured. The subject matter is very attractive. Endocrine surgery is widely practiced in all departments of general surgery. Experience and authoritativeness of the author increase the value of the manuscript.

Author Response

No comments. Thanks for your appraisal.

Reviewer 3 Report

I have found no clear link between references 41 and the content
of the author's sentence. Perhaps this can be explained.
Primary hyperparathyroidism should lead to suppression
of healthy parathyroid glands. And the article concerns thyroidectomy
and normal parathyroid glands, not hyperfunctioning.
***
I have doubts about the term parathyroid splinting.
Nevertheless, I am not a linguist and the author,
when creating a hypothesis, has the right to use his own nomenclature.
***
In one of the references (78), the author cites his hypothesis.
***
This is a very interesting review based on the author's own
and other specialists' experiences.
Author points to an important clinical problem
which are disorders of parathyroid function after thyroidectomy.
Attacks stereotypes about autotransplantation of the parathyroid glands
including those showing the features of ischemia.
Formulates a hypothesis which, however, requires confirmation
in prospective studies, what should be clearly indicated.
The article shows the importance of the surgeon's approach
to preserving the parathyroid glands in their original location,
which may come as a surprise to many surgeons.
Discussion is to be expected.

Author Response

1) Reference 41 supports the clinical evidence that when thyroidectomy is associated to parathyroidectomy for hyperparathyroidism, the chances of hypoparathyroidism increase because at leas one gland is missing. I do not modifications are necessary.

2) Splinting is a widely used term indicating support of a fractured bone, for example. It was first used by the author (ref. 14) to suggest that the failing parathyroid gland should be "splinted", as we do with fractured bones or damaged organs, by keeping the serum calcium in the normal-high range.

3) I agree with the reviewer that this is a thought-provoking article introducing several new concepts that will capture the interest of surgeoons and endocrinologists interested in post-surgical hypoparathyroidism.

I do not think ammendements are necessary

Reviewer 4 Report

In the manuscript " Etiology and diagnosis of permanent hypoparathyroidism after total thyroidectomy", prof. Antonio Sitges-Serra  performed a literature review on postoperative hypoparathyroidism after total thyroidectomy.  The paper is well written and has important clinical significance. The paper highlights a major issue in the field of surgical endocrinology and makes important  and usefull remarks.

  • Minor English revision.

In the manuscript “Etiology and diagnosis of permanent hypoparathyroidism after total thyroidectomy”, the author writes an extensive and interesting review on literature studies.

The author present also a subjective point of view on the operative outcomes and techniques on preserving parathyroid glands in total thyroidectomy, and I this this should be mentioned, as well as the need to verify his hypothesis through clinical studies.

I find the paper with important clinical values, as it makes important points in postoperative hyperparathyroidism, as it is present in a significant percentage after total thyroidectomy, however a methodology should be presented on how and based on what consideration he selected the literature article for review.

I did not find any structural flaws, as I find well written. A short discussion on interventions concerning primary and hyperparathyroidism should be interesting, as it could present a surgeon’s perspective on how to manage the preoperative and postoperative treatment.

A short table shortly describing the stages of evolving iatrogenic states could make the lecture easier.

Author Response

No comments. Thanks to the reviewer for his/her positive appraisal.

This last reviewer has certainly grasped the innovative scope and the fresh blood my article brings to the field. Thanks.

1) I politely disagree with your reviewer about the concept of “subjective point of view” since no affirmation is made along the paper without some supportive evidence. It is not a speculative paper although, I agree with your reviewer, author’s claims need to be supported by further studies in other units. Most of the data presented –either from the author’s unit or from other experts- are based on careful and protocol-guided retrospective and prospective clinical observations.

2) I have modified the last paragraph of the Introduction to clarify the basis for selecting the most relevant literature on the topic.

3) I have added a new section on hypopara after parathyroidectomy at the near-end of the manuscript

4) A new figure (#2) has been added to clarify the different phases of postoperative hypoparathyroidism

Reviewer 5 Report

Excellently written

Round 2

Reviewer 1 Report

In my opinion, for this theme would been better a systematic review, however as narrative review the article is acceptable in this form.